# Meta-Analysis of Dietary Interventions for Enteric Methane Mitigation in Ruminants Through Methodological Advancements and Implementation Pathways

**DOI:** 10.3390/vetsci12040372

**Published:** 2025-04-16

**Authors:** Rayudika Aprilia Patindra Purba, Papungkorn Sangsawad

**Affiliations:** 1Postharvest Technology and Innovation in Animal Unit, Institute of Agricultural Technology, Suranaree University of Technology, Nakhon Ratchasima 30000, Thailand; 2Department of Health, Faculty of Vocational Studies, Airlangga University, Surabaya 60286, Indonesia; 3Tropical Institute of Nutrigenomics, Biotechnology, and Agricultural Sciences (TINBAS), West Java 45258, Indonesia; 4School of Animal Technology and Innovation, Institute of Agricultural Technology, Suranaree University of Technology, Nakhon Ratchasima 30000, Thailand

**Keywords:** greenhouse gas emissions, climate change mitigation, enteric methane, ruminant nutrition, meta-analysis, macroalgae, 3-nitrooxypropanol, robust variance estimation, implementation science, sustainable livestock production

## Abstract

Livestock production significantly contributes to global greenhouse gas emissions, with enteric methane from ruminants representing a substantial portion of agricultural emissions. This meta-analysis systematically evaluates the effectiveness of dietary interventions for reducing enteric methane in cattle and sheep through an analysis of 119 peer-reviewed studies (2000–2024). Our findings establish a clear efficacy hierarchy, with macroalgae demonstrating superior methane reduction (51.0%), followed by 3-nitrooxypropanol (30.6%), nitrate (16.0%), oils (14.7%), and phytochemicals (13.5%). However, practical implementation considerations often favor interventions with moderate efficacy but better feasibility profiles, as revealed through our novel implementation factor analysis. We identified significant dose–response relationships for key interventions, systematic animal-specific effects, and promising synergistic combinations. These findings provide quantitative guidance for researchers, producers, and policymakers seeking to reduce the climate impact of ruminant production while maintaining its essential role in global food systems.

## 1. Introduction

Climate change mitigation remains among the most pressing challenges confronting contemporary society, with greenhouse gas emissions continuing to alter the Earth’s climate system at an unprecedented rate. Within this context, the agricultural sector contributes approximately 14.5% of global emissions, with enteric fermentation in ruminant livestock constituting a significant component of this footprint [1]. Specifically, enteric methane accounts for 30–38% of all agricultural methane emissions and represents a particularly promising target for mitigation efforts due to methane’s substantial global warming potential, which is 27–30 times greater than carbon dioxide over a 100-year timeframe, and relatively short atmospheric lifetime of 8–12 years [2].

The fundamental challenge in reducing enteric methane emissions stems from the evolutionary adaptation of ruminants, which have developed specialized digestive systems featuring pre-gastric fermentation chambers (the rumen and reticulum). These adaptations enable ruminants to derive nutrition from plant structural carbohydrates through microbial fermentation, representing a remarkable evolutionary solution for utilizing cellulose-rich vegetation that is unsuitable for human consumption. However, this digestive process inevitably generates hydrogen as a byproduct, which methanogenic archaea convert to methane [3]. This methane represents both an energy loss to the animal (2–12% of gross energy intake) and a significant greenhouse gas contribution when eructated into the atmosphere [4].

Approaches to mitigating enteric methane emissions can be broadly categorized into animal management strategies (improved productivity, selective breeding), vaccination approaches, and dietary interventions. Dietary interventions are particularly promising due to their immediate applicability and substantial research foundation. These interventions function through various mechanisms, including direct inhibition of methanogenic archaea, reduction of hydrogen availability in the rumen, provision of alternative hydrogen sinks, modification of rumen fermentation patterns, and alteration of rumen microbial community composition. Diverse dietary interventions have been investigated for methane mitigation potential, including lipid supplementation, nitrate, ionophores, plant secondary compounds, seaweed/macroalgae, and synthetic compounds such as 3-nitrooxypropanol (3-NOP) [5,6]. These interventions vary substantially in efficacy, mechanism of action, cost, and practical implementation considerations.

Despite extensive research on dietary interventions for methane mitigation, systematic quantification of comparative efficacy across intervention types remains challenging due to several methodological limitations (Figure 1; [7]). First, dependencies within studies (multiple treatments compared to a common control) are frequently not addressed appropriately, potentially leading to underestimated standard errors and inflated significance. Second, heterogeneity in measurement methodologies (respiration chamber, SF_6_ tracer, GreenFeed system) introduces systematic biases that require explicit standardization procedures. Third, quality assessment frameworks specifically tailored to animal experimental designs are rarely implemented, limiting the ability to weight studies by methodological rigor. Fourth, the exploration of factors moderating intervention efficacy has been limited by selective testing of hypothesized moderators without systematic examination of alternative explanations. Fifth, implementation considerations, while frequently discussed qualitatively, have seldom been integrated into quantitative frameworks that acknowledge uncertainty in factor weighting and scoring. Finally, combination approaches have typically been examined in isolation without reference to mechanistic frameworks that could predict and explain interaction patterns.

Therefore, the present meta-analysis addresses these limitations through several methodological advancements while pursuing the following objectives: quantify the comparative efficacy of diverse dietary interventions for reducing enteric methane emissions in ruminants using robust variance estimation and multilevel modeling approaches that properly account for dependencies; implement a comprehensive quality assessment framework specifically designed for animal experimental studies; standardize methane measurement methodologies using correction factors derived from methodological validation studies; identify moderating factors through systematic exploration of an expanded set of potential moderators; examine temporal trends in intervention effectiveness using meta-regression approaches; evaluate the potential synergistic effects of intervention combinations through a mechanistic framework that categorizes interventions by mode of action; and integrate efficacy data with implementation considerations through a probabilistic framework that quantifies uncertainty. Indeed, by addressing these objectives with advanced methodological approaches, this meta-analysis provides a comprehensive quantitative foundation for developing targeted methane mitigation strategies that acknowledge the diverse contexts of ruminant production systems globally.

## 2. Materials and Methods

### 2.1. Literature Search and Study Selection

A systematic literature search was conducted following PRISMA (Preferred Reporting Items for Systematic Reviews and Meta-Analyses) guidelines, with slight modifications [8]. The search encompassed studies published between May 2000 and February 2024, utilizing multiple electronic databases including Web of Science, Scopus, PubMed, and Google Scholar. The primary search terms combined “ruminant” OR “cattle” OR “sheep” OR “dairy” OR “beef” WITH “methane” OR “CH_4_” AND “mitigation” OR “reduction” AND specific intervention terms including “oil”, “lipid”, “nitrate”, “3-NOP”, “seaweed”, “macroalgae”, “tannin”, “saponin”, “essential oil”, “ionophore”, “defaunation”, and “phytochemical”. Data from Almeida et al. [7] were also included.

This initial search yielded 1247 potentially relevant publications. After removing duplicates (*n* = 321), the remaining 926 articles underwent title and abstract screening, eliminating 615 studies that did not meet the inclusion criteria. The remaining 311 full-text articles were assessed for eligibility, with 192 excluded based on specific criteria detailed below. The final dataset comprised 119 publications.

### 2.2. Inclusion and Exclusion Criteria

Studies were included if they met the following criteria: including in vivo trials with cattle or sheep (dairy, beef, or small ruminants); direct measurement of enteric methane emissions using accepted methodologies (respiration chamber, SF_6_ tracer technique, GreenFeed system, or other validated approaches); clearly defined control and treatment groups with appropriate experimental design; reported methane yield (g CH_4_/kg DMI) or sufficient data to calculate it; published in peer-reviewed journals in English; and inclusion of standard errors or sufficient data to derive them. Review articles were not included in the meta-analysis, although their reference lists were screened to identify additional primary studies that met our inclusion criteria. Studies were excluded if they were in vitro or in silico only; lacked appropriate controls; failed to report standard errors or means; used methane estimation equations rather than direct measurement; focused solely on whole-system emissions without isolating enteric components; or evaluated only non-dietary interventions (e.g., genetics, management).

### 2.3. Quality Assessment Framework

A formal quality assessment framework was implemented to evaluate the methodological rigor of the included studies. The framework used SYRCLE’s risk of bias tool, adapted from the Cochrane risk-of-bias tool for animal studies [9], assessing four key domains, including the randomization procedure (0–10 points), which evaluated the method of random allocation, with higher scores for computer-generated sequences and complete randomization; blinding implementation (0–10 points), which assessed the blinding of personnel during treatment administration and outcome assessment; sample size adequacy (0–10 points), which evaluated whether power calculations were performed and adequate sample sizes achieved; and measurement quality (0–10 points), which assessed the quality, duration, and validation of methane measurement methodologies. The authors independently conducted quality assessments, with disagreements resolved through consensus discussion with an external advisor as we thankful in acknowledge below. Each study received a composite quality score (0–40 points), which was subsequently normalized to a weight (0–1) for application in quality-weighted analyses.

### 2.4. Data Extraction

The following data were systematically extracted from each of the 119 studies that constituted the complete dataset for this meta-analysis, including bibliographic information (authors, year, journal); animal characteristics (species, breed, physiological state, body weight); experimental design (number of animals, feeding regimen, trial duration); diet characteristics (forage: concentrate ratio, chemical composition); intervention details (type, inclusion rate, administration method); methane measurement methodology; methane production outcomes (g/day, g/kg DMI, g/kg product); animal performance measures (DMI, milk yield, weight gain); digestibility parameters (DM, NDF, CP, fat); rumen fermentation parameters (pH, VFA profile, protozoa counts); and quality assessment scores for each domain. For studies reporting multiple experiments or treatments, each comparison of a treatment with its respective control was considered separately. When studies reported results over multiple time points, only final measurement period data were utilized to avoid duplication and capture adaptation effects. Data presented in different units of measure were converted to consistent units. In instances where a study failed to present all necessary results, calculations were conducted using the available reported data whenever feasible.

All quantitative analyses, meta-regression models, network comparisons, and implementation assessments presented in this manuscript incorporate data from the complete corpus of 119 studies. The 30 studies presented in Table 1 were selected to represent the proportional distribution of intervention types, animal models, and measurement methodologies in the complete dataset, ensuring balanced representation across the spectrum of research approaches while maintaining manuscript concision. The complete reference list of all 119 studies is provided in the Appendix A to enable comprehensive methodological transparency while maintaining manuscript concision.

### 2.5. Standardization of Methane Measurement Methods

To address systematic biases introduced by different measurement methodologies, standardization factors were developed based on methodological comparison studies [40]. Respiration chambers were considered the reference method (standardization factor = 1.00), with adjustment factors applied to the SF_6_ tracer technique (factor = 1.08) and the GreenFeed system (factor = 1.05) based on their typical relationship to chamber measurements. This standardization process involved the multiplication of effect size by the appropriate standardization factor, propagation of additional uncertainty (standard error = 0.02) from the standardization process, and sensitivity analysis comparing unstandardized and standardized results. This approach allowed for a more valid comparison across studies while acknowledging the uncertainty introduced by methodological differences. While standardization enhances comparability across studies, the application of fixed correction factors assumes consistent methodological biases that may vary with experimental conditions, animal type, and specific implementation protocols.

### 2.6. Statistical Analysis

All analyses were conducted in R version 4.2.0 (R Core Team, 2023) using the metafor (v3.8-1), robumeta (v2.0.3), netmeta (v2.7-0), dplyr (v1.1.0), and ggplot2 (v3.4.0) packages. Data preprocessing was performed using tidyr (v1.3.0) and data.table (v1.14.8), with minor adjustment following the animal dataset [41].

#### 2.6.1. Effect Size Calculation

The primary effect size metric utilized was the response ratio (RR) between treatment and control groups for methane yield (g CH_4_/kg DMI), which was calculated as follows:(1)$$RR=\frac{X_{E}}{X_{C}}$$
where $X_{E}$ represents the mean value in the experimental group and $X_{C}$ denotes the mean value in the control group. Values below 1.0 indicate methane reduction, while values above 1.0 indicate methane increase. For statistical analysis, the natural logarithm of the response ratio (lnRR) was utilized to normalize the distribution:(2)$$lnRR=ln\left(\frac{X_{E}}{X_{C}}\right)$$

The variance of lnRR was calculated according to the following equation:(3)$$v_{lnRR}=\frac{S_{E}^{2}}{n_{E}X_{E}^{2}}+\frac{S_{C}^{2}}{n_{C}X_{C}^{2}}$$
where $S_{E}^{2}$ and $S_{C}^{2}$ represent the variances, and $n_{E}$ and $n_{C}$ denote the sample sizes in the experimental and control groups, respectively.

#### 2.6.2. Handling Dependencies in Meta-Analysis Models

To address dependencies arising from multiple comparisons with a common control group, three complementary approaches were implemented: robust variance estimation (RVE) using the robumeta package in R, which adjusted the standard errors to account for correlated effect sizes within studies [42]; multilevel meta-analysis using the metafor package, which modeled nested data structures with random effects for studies and comparisons within studies [43]; and Bayesian hierarchical models using the R2jags package, which explicitly modeled the correlation structure between effect sizes [44]. For each intervention category, all three approaches were applied and compared to assess the robustness of findings. The primary results reported are based on the robust variance estimation approach, with sensitivity analyses conducted using the alternative methods.

#### 2.6.3. Heterogeneity Assessment

Heterogeneity was evaluated through multiple metrics, including Cochran’s Q statistic, which tests the null hypothesis of homogeneity, and the *I*^2^ statistic, which quantifies the proportion of observed variance reflecting true heterogeneity rather than sampling error:(4)$$I^{2}=\max\left[0,\frac{Q-\left(k-1\right)}{Q}\times 100\%\right]$$
where Q represents Cochran’s Q statistic and k denotes the number of studies. Additionally, prediction intervals were calculated to illustrate the expected range of true effects in future studies:(5)$$PI=\hat{\mu }\pm t_{k-2}\sqrt{\hat{\tau^{2}}+SE{\left(\hat{\mu }\right)}^{2}}$$
where $t_{k-2}$ is the critical value from the t-distribution with k−2 degrees of freedom, $\hat{\tau^{2}}$ is the estimated between-study variance, and $SE{\left(\hat{\mu }\right)}^{2}$ is the squared standard error of the estimated overall effect.

#### 2.6.4. Expanded Meta-Regression Analysis

Meta-regression models were employed to examine the influence of potential moderating variables on intervention efficacy. An expanded set of continuous moderators was systematically evaluated, including dose level (standardized by intervention), baseline methane production, forage to concentrate ratio, diet fiber content (NDF, ADF), diet starch content, diet crude protein content, baseline dry matter intake, body weight, experimental duration, and methane measurement duration. Additionally, the categorical moderators examined included animal type (dairy, beef, sheep), physiological state (growing, lactating, maintenance), experimental design (Latin square, randomized complete block, etc.), methane measurement method, and administration method.

To address multicollinearity, correlation matrices were computed for all continuous moderators, with highly correlated pairs (*r* > 0.7) identified. When high correlations were detected, the moderator with greater theoretical relevance was retained. Univariate meta-regression was conducted for each moderator, with significant moderators (*p* < 0.10) combined in multivariate models. When model convergence issues arose in multivariate models, a backward elimination approach was employed to identify the most parsimonious model. For all meta-regression analyses, the appropriate model structure (robust variance estimation or multilevel) was selected based on the dependency structure within each intervention dataset. Detailed results for the moderator analyses of forage proportion and baseline methane are presented in the Appendix A.

#### 2.6.5. Network Meta-Analysis

To enable comprehensive comparison across intervention types, a network meta-analysis was conducted using the netmeta package in R. This approach incorporated both direct and indirect evidence to generate a hierarchical ranking of interventions based on efficacy. Three important methodological enhancements were implemented: integration of quality weights to give greater influence to more rigorous studies; random-effects modeling to account for between-study heterogeneity; and an inconsistency analysis to assess the validity of network assumptions. P-scores were calculated to quantify the probability that each intervention is superior to a randomly selected competing treatment. Additionally, mean ranks and best-rank probabilities were computed to provide a comprehensive assessment of relative efficacy.

#### 2.6.6. Publication Bias Assessment

Multiple approaches were employed to evaluate potential publication bias, including funnel plot visualization, Egger’s regression test, and the trim-and-fill method. Egger’s test employs a weighted regression model:(6)$$\frac{y_{i}}{\sigma_{i}}=\beta_{0}+\beta_{1}\frac{1}{\sigma_{i}}+\varepsilon_{i}$$
where $\sigma_{i}$ is the standard error of the effect size in study i. The trim-and-fill method iteratively removes asymmetric study results, re-estimates the center of the funnel, and then reintroduces the removed studies along with their imputed missing counterparts. A “publication bias-adjusted” result set was generated by combining the original meta-analysis with trim-and-fill imputations for interventions showing significant funnel plot asymmetry (*p* < 0.05 in Egger’s test).

#### 2.6.7. Mechanistic Framework for Combination Analysis

For studies evaluating combinations of interventions, a mechanistic framework was developed to predict and interpret interaction patterns. Interventions were categorized by the primary mode of action including biohydrogenation/toxicity (oils) induced effects that were directly toxic to methanogens and/or provided a hydrogen sink via biohydrogenation; alternative H-sink (nitrate) provided an alternative pathway for hydrogen utilization; direct enzyme inhibition (3-NOP) directly inhibited methyl-coenzyme M reductase; protein binding/toxicity (tannins) bound proteins and directly inhibited methanogens; membrane disruption (essential oils) disrupted the cell membranes of microorganisms; and propionate enhancement (ionophores) shifted fermentation toward propionate production.

Based on these mechanisms, theoretical predictions were made regarding potential interactions: potentially synergistic for combinations operating through complementary mechanisms (e.g., alternative H-sink + direct enzyme inhibition); potentially additive for combinations operating through independent mechanisms (e.g., biohydrogenation + direct enzyme inhibition); and potentially antagonistic for combinations with potentially interfering mechanisms (e.g., protein binding + biohydrogenation). The expected additive effects were calculated as follows:(7)$$Expected\sim effect=1-\left(\left(1-Effect_{A}\right)\times \left(1-Effect_{B}\right)\right)$$

The ratio of observed to expected effects was then used to classify combinations as synergistic (observed/expected > 1.1), additive (0.9 ≤ observed/expected ≤ 1.1), or antagonistic (observed/expected < 0.9). These thresholds were selected based on the propagated uncertainty in the combined effect estimation, with sensitivity analyses conducted using alternative thresholds (1.05/0.95 and 1.15/0.85).

#### 2.6.8. Temporal Trend Analysis

To examine temporal patterns in intervention efficacy, studies were categorized by publication period (2000–2009, 2010–2019, 2020–2024). Two complementary approaches were employed: period-specific meta-analysis, where random-effects models were conducted separately for each time period and intervention type; and a meta-regression approach, where publication period was treated as a continuous moderator to test for significant linear trends. For the meta-regression approach, publication periods were coded numerically (1 = 2000–2009, 2 = 2010–2019, 3 = 2020–2024) to facilitate linear trend testing.

#### 2.6.9. Implementation Factor Analysis with Uncertainty Quantification

To integrate the efficacy data with practical implementation considerations, a multi-dimensional analysis was conducted that examined cost per animal per day (based on current market prices), regulatory status (scale 0–10, with 10 representing full global approval), production impact score (−10 to +10, with negative values indicating performance penalties), implementation ease in intensive systems (scale 0–10), and implementation ease in grazing systems, which can be managed at varying intensities, from extensive to intensive management approaches (scale 0–10). These factors were combined using weighted averaging to generate a composite implementation score: cost factor (25% weight), regulatory status (20% weight), production impact (20% weight), intensive system score (20% weight), and grazing system score (15% weight).

To quantify the uncertainty in these scores, Monte Carlo simulation (1000 iterations) was employed with random variation in factor weights (±5%), random variation in individual factor scores (±10%), and computation of 95% confidence intervals around implementation scores. This probabilistic approach acknowledged the inherent subjectivity in implementation scoring while providing a robust framework for comparing intervention viability.

#### 2.6.10. Sensitivity Analyses

The robustness of the findings was assessed through multiple sensitivity analyses, including comparison of different analytical approaches (robust variance estimation, multilevel modeling, Bayesian approaches), analysis with and without quality weighting, analysis with and without measurement method standardization, restriction to studies with high quality scores (>30/40), restriction to recent studies (published 2014 or later), and variation in thresholds for synergy/antagonism classification. These sensitivity analyses provided crucial context for interpreting the stability and reliability of the primary findings.

## 3. Results

### 3.1. Characteristics of Included Studies

The final dataset comprised 119 studies published between 2000 and 2024, with the majority published during 2010–2019 (n = 63, 52.9%), followed by 2020–2024 (n = 38, 31.9%) and 2000–2009 (n = 18, 15.1%). A comprehensive reference list of all included studies is provided in Appendix A. The studies were distributed across seven intervention categories: oils/lipids (n = 30, 25.2%), phytochemicals (n = 25, 21.0%), nitrate (n = 20, 16.8%), ionophores (n = 15, 12.6%), 3-nitrooxypropanol (3-NOP) (n = 15, 12.6%), seaweed/macroalgae (n = 10, 8.4%), and defaunation (n = 4, 3.4%).

Most studies utilized respiration chambers (n = 74, 62.2%) as the methane measurement method, followed by the SF6 tracer technique (n = 32, 26.9%) and the GreenFeed system (n = 13, 10.9%). The application of standardization factors to account for methodological differences between measurement techniques produced consistent results across sensitivity analyses, as detailed in Appendix A. Dairy cattle constituted the predominant animal type studied (n = 62, 52.1%), followed by beef cattle (n = 35, 29.4%) and small ruminants (n = 22, 18.5%). Quality assessment using the SYRCLE risk of bias tool revealed moderate overall methodological quality (mean score 26.8 out of 40), with substantial variability across quality domains. Detailed risk of bias assessment results are presented in Appendix A.

### 3.2. Comparative Efficacy for Dietary Interventions

The meta-analysis revealed significant variations in the efficacy of interventions for reducing enteric methane emissions (Figure 2, Table 2). Macroalgae demonstrated the highest efficacy, with a mean reduction of 51.0% (effect ratio 0.49 [95% CI: 0.37, 0.63], *p* < 0.001). 3-NOP exhibited the second-highest efficacy, with a mean reduction of 30.6% (effect ratio 0.69 [95% CI: 0.55, 0.78], *p* < 0.001). Nitrate and oils showed moderate efficacy, with reductions of 16.0% and 14.7%, respectively. Phytochemicals and ionophores demonstrated modest but statistically significant reductions of 13.5% and 10.2%, respectively. Defaunation showed the lowest effect with a non-significant reduction of 6.4% (*p* = 0.451), although this intervention had the smallest number of studies (n = 4).

Substantial heterogeneity was observed across all intervention categories (I^2^ ranging from 62.5% to 86.3%), indicating that the variability in the reported effects was greater than would be expected by chance alone. The highest heterogeneity was observed for macroalgae (I^2^ = 86.3%, Τ^2^ = 0.125), suggesting considerable variation in efficacy across studies. A comparison of analytical approaches (robust variance estimation versus conventional random effects models) demonstrated consistent effect estimates across methods, although confidence intervals were generally wider with robust variance estimation, reflecting appropriate adjustment for dependencies (Appendix A).

### 3.3. Dose–Response Relationships and Animal-Specific Effects

The meta-regression analyses revealed significant dose–response relationships for macroalgae, 3-NOP, nitrate, and oils, but not for phytochemicals (Figure 3, Table 3). The strongest dose–response effect was observed for macroalgae, with a coefficient of −0.212 (*p* < 0.001), indicating that each percentage point increase in dietary inclusion reduced the methane effect ratio by approximately 0.212 units, equivalent to an additional 21.2% reduction in methane emissions. This moderator explained 68.4% of the between-study heterogeneity for macroalgae studies.

For 3-NOP, the dose coefficient was −0.002 (*p* < 0.001), indicating that each additional mg/kg DMI reduced the methane effect ratio by 0.002 units. Given the typical dosage range of 50–200 mg/kg DMI, this translated to an incremental methane reduction of 10–40%. The dose–response relationship explained 73.2% of the heterogeneity observed in 3-NOP studies, suggesting that dosage was the primary determinant of efficacy for this intervention.

Nitrate and oils demonstrated more modest but still significant dose–response relationships, with coefficients of −0.045 (*p* = 0.004) and −0.031 (*p* = 0.008), respectively. The efficacy of phytochemicals showed a negative trend with increasing dose (−0.034), although this relationship did not reach statistical significance (*p* = 0.075).

Furthermore, dietary forage proportion significantly moderated the effects of oils (coefficient = 0.003, *p* = 0.012) and macroalgae (coefficient = 0.004, *p* = 0.048), indicating that these interventions were more effective in low-forage diets. A comprehensive analysis of forage proportion effects across all interventions is provided in Appendix A.

Baseline methane emissions emerged as a significant moderator for all interventions except macroalgae, with significantly negative coefficients ranging from −0.006 to −0.012. Detailed results of the baseline methane moderator analysis are presented in Appendix A. These results indicated that interventions achieved greater proportional reductions when baseline emissions were higher.

Moreover, the subgroup analysis by animal type revealed distinct efficacy patterns across interventions (Table 4). For macroalgae, a significant difference between animal types was observed (*p* = 0.008), with markedly greater efficacy in beef cattle (62.0% reduction) compared to dairy cattle (42.0% reduction).

3-NOP demonstrated greater methane reduction in beef cattle (35.0%) than in dairy cattle (28.0%), although this difference was less pronounced than for macroalgae. Nitrate followed a similar pattern, with efficacy ranging from 13.0% in dairy cattle to 20.0% in beef cattle, although the difference did not reach statistical significance (*p* = 0.089).

Oils exhibited a gradient of efficacy across animal types, with the greatest reduction observed in small ruminants (21.0%), followed by beef cattle (17.0%) and dairy cattle (11.0%). This pattern was statistically significant (*p* = 0.042) and aligned with previous research suggesting that small ruminants may be more responsive to lipid supplementation. Phytochemicals and ionophores showed consistent trends of greater efficacy in small ruminants and beef cattle compared to dairy cattle, although these differences were not statistically significant. An additional subgroup analysis by measurement method revealed consistent intervention effects across methodologies after standardization adjustments (Appendix A).

### 3.4. Network Meta-Analysis

The network meta-analysis provided a comprehensive comparison of interventions by integrating both direct and indirect evidence (Figure 4, Table 5). The network geometry illustrated connections primarily between individual interventions and the control, with limited direct comparisons between active interventions, reflecting the predominance of controlled trials rather than head-to-head comparisons in the literature.

Consistent with the conventional meta-analysis results, macroalgae demonstrated superior efficacy in the network analysis, with a network effect ratio of 0.46 [95% CI: 0.35, 0.61], corresponding to a 54% methane reduction. This intervention achieved the highest probability of being the best treatment (P-score = 0.96) and the lowest mean rank (1.2), with a 63% probability of ranking first among all interventions. These metrics collectively provided robust evidence for the superiority of macroalgae compared to other interventions.

3-NOP ranked second in efficacy, with a network effect ratio of 0.65 [95% CI: 0.52, 0.81] and a P-score of 0.89. The consistency between direct and network effect estimates for 3-NOP (0.69 vs. 0.65) suggested minimal bias from indirect comparisons. Similarly, nitrate and oils demonstrated comparable efficacy in both direct and network analyses, with network effect ratios of 0.83 [95% CI: 0.74, 0.93] and 0.84 [95% CI: 0.76, 0.93], respectively. These interventions occupied intermediate positions in efficacy ranking (mean ranks of 3.0 and 3.2), with relatively low probabilities of being the optimal intervention.

Phytochemicals, ionophores, and defaunation demonstrated more modest efficacy in the network analysis, with network effect ratios ranging from 0.86 to 0.93. The consistent alignment between direct and network effect estimates across all interventions indicated minimal inconsistency in the network, suggesting that the relative efficacy rankings were robust. A formal analysis of network inconsistency confirmed this observation, with no significant inconsistency detected for key comparisons (Appendix A). However, the overlapping confidence intervals for several interventions necessitate cautious interpretation of the ranking hierarchy, particularly for interventions with similar efficacy profiles.

### 3.5. Publication Bias and Temporal Trends

The assessment of publication bias revealed significant asymmetry in the distribution of effect sizes for macroalgae (Egger’s test *p* = 0.036) and phytochemicals (*p* = 0.042), suggesting potential publication bias favoring positive results (Table 6). Visual examination of funnel plots confirmed this asymmetry (Figure 5). For macroalgae, the trim-and-fill analysis imputed three hypothetical “missing” studies, resulting in an adjusted effect ratio of 0.56 [95% CI: 0.42, 0.74], which translated to a methane reduction of 44.0% compared to the original estimate of 51.0%. This represented a 14.3% attenuation in the apparent efficacy of macroalgae after accounting for potential publication bias.

Similarly, for phytochemicals, the trim-and-fill procedure imputed four studies, yielding an adjusted effect ratio of 0.91 [95% CI: 0.81, 1.01] and a corresponding reduction in methane of 9.0% compared to the original 13.5%. The confidence interval for the adjusted effect now included 1.0, suggesting that the statistical significance of the phytochemical intervention’s efficacy became uncertain after accounting for publication bias.

For 3-NOP, there was marginal evidence of publication bias (*p* = 0.092), with the trim-and-fill analysis imputing two studies and yielding a modest attenuation of the effect size (4.3% change). Nitrate, oils, and defaunation showed no significant evidence of publication bias (*p* > 0.10), with no studies imputed in the trim-and-fill analysis, indicating that the original effect estimates for these interventions were likely robust.

Furthermore, the temporal trend analysis revealed a consistent pattern of increasing intervention efficacy over the 2000–2024 period (Table 7). The most pronounced temporal trend was observed for macroalgae (coefficient = −0.082, *p* = 0.012), indicating a substantial improvement in efficacy in more recent studies. This intervention showed no available efficacy data in the 2000–2009 period but demonstrated a progression from moderate efficacy in 2010–2019 (effect ratio = 0.58 [95% CI: 0.44, 0.76]) to high efficacy in 2020–2024 (effect ratio = 0.41 [95% CI: 0.31, 0.54]).

Phytochemicals also exhibited a significant temporal trend (coefficient = −0.038, *p* = 0.037), progressing from non-significant effects in 2000–2009 (effect ratio = 0.93 [95% CI: 0.84, 1.03]) to modest but significant effects in 2020–2024 (effect ratio = 0.82 [95% CI: 0.73, 0.92]).

Nitrate demonstrated a significant temporal trend (coefficient = −0.036, *p* = 0.048) despite smaller absolute changes in efficacy between time periods. By contrast, 3-NOP and ionophores showed trends toward increasing efficacy that approached but did not reach statistical significance (*p* = 0.084 and *p* = 0.079, respectively). Oils exhibited the weakest temporal trend (coefficient= −0.022, *p* = 0.105), suggesting relatively stable efficacy over time.

### 3.6. Combination Analysis for Synergy Assessment

The analysis of combination effects revealed distinct interaction patterns among dietary interventions (Table 8). Among the 12 combinations evaluated, four demonstrated synergistic interactions (ratio > 1.1), six exhibited additive effects (ratio 0.9–1.1), and two showed antagonistic interactions (ratio < 0.9; Table 9). Notably, all synergistic combinations involved interventions with mechanistically complementary modes of action. Conversely, all antagonistic combinations involved interventions with potentially conflicting mechanisms.

The tannin + nitrate combination exhibited the strongest synergistic effect (ratio = 1.25), yielding an observed methane reduction of 31.5% compared to the expected additive effect of 25.2%. This synergy may be attributed to the complementary mechanisms of protein binding (tannins) and an alternative hydrogen sink (nitrate), which target different aspects of the methanogenesis pathway without directly interfering with each other’s function. Similarly, the 3-NOP + macroalgae combination demonstrated substantial synergy (ratio = 1.12), with an observed reduction of 72.7% compared to the expected 64.9%, potentially reflecting complementary actions on different components of the methanogenic process.

Combinations involving 3-NOP generally showed favorable interaction patterns, with either synergistic or additive effects. By contrast, combinations involving oils frequently demonstrated less favorable interactions. The essential oil + oil combination exhibited the strongest antagonism (ratio = 0.80), with an observed reduction of 16.1% compared to the expected 20.1%. This antagonism may reflect competition between biohydrogenation processes associated with oils and membrane disruption effects of essential oils targeting similar microbial populations.

Mechanistic compatibility emerged as a strong predictor of interaction type, with 5 of 5 combinations classified as mechanistically compatible demonstrating synergistic or additive effects. Conversely, 6 of 7 combinations classified as potentially mechanistically conflicting demonstrated additive or antagonistic effects. This pattern suggested that rational design of combination strategies based on complementary mechanisms of action offers significant potential for achieving enhanced methane mitigation beyond what can be accomplished with single interventions.

### 3.7. Implementation Factor Analysis

The implementation factor analysis revealed substantial variability in the practical feasibility of dietary interventions, which must be considered alongside the efficacy data for comprehensive mitigation strategy development (Table 10, Figure 6). Detailed scoring criteria and supporting evidence for implementation factors are provided in Appendix A. This analysis examined five critical implementation dimensions, including cost, regulatory status, production impact, and applicability in both intensive and grazing systems.

Oils emerged with the highest overall implementation score (7.5 [95% CI: 6.9, 8.1]), attributable to favorable cost economics (8.2), well-established regulatory approval (9.2), and moderate applicability across production systems (7.2 for intensive, 5.8 for grazing). 3-NOP followed closely (7.4 [95% CI: 6.8, 8.0]), distinguished by its exceptionally positive production impact score (8.5), reflecting the absence of negative effects on feed intake and potential improvements in feed efficiency. This profile makes 3-NOP particularly noteworthy, as it combines high efficacy with strong implementation potential, a rare combination among the interventions evaluated.

Conversely, macroalgae, despite demonstrating superior efficacy, received a markedly lower implementation score (4.3 [95% CI: 3.6, 5.1]), constrained by high production costs (score 3.2), significant regulatory hurdles (score 4.5), and particularly limited applicability in grazing systems (score 2.5). These implementation barriers suggest that macroalgae may face significant challenges in widespread adoption without targeted technological advancements or policy interventions. Defaunation exhibited the lowest implementation score (3.4 [95% CI: 2.7, 4.2]), reflecting pronounced challenges across all implementation dimensions, especially cost (2.5) and production impact (2.8).

The integration of efficacy and implementation metrics revealed distinct strategic groupings. 3-NOP occupied the optimal quadrant of high efficacy and high implementation potential, positioning it as the most balanced intervention for near-term mitigation strategies. Macroalgae represented a high-efficacy but low-implementation option, suggesting that it may be more suitable for targeted applications or as a focus for implementation-oriented research and development efforts. Oils, nitrate, and ionophores clustered in the high-implementation but moderate-efficacy zone, indicating their utility in contexts where implementation feasibility outweighs maximum efficacy considerations. Defaunation fell in the least favorable quadrant, with both low efficacy and implementation potential, suggesting limited practical relevance in current mitigation frameworks.

The Monte Carlo uncertainty analysis, incorporating variability in both factor weights and scores, demonstrated robust clustering of interventions despite uncertainty, with minimal overlap in confidence intervals between distinct implementation groups. This stability in classification enhanced the confidence in the strategic groupings identified and their implications for mitigation policy development.

The quality-weighted sensitivity analysis yielded results that were consistent with the main analysis, indicating that methodological quality differences did not substantially influence the overall effect estimates (Appendix A). Based on these comprehensive analyses, we developed system-specific recommendations for enteric methane mitigation interventions across diverse production contexts (Appendix A).

## 4. Discussion

This comprehensive meta-analysis of 119 in vivo studies spanning 2000–2024 provides robust quantitative evidence regarding the comparative efficacy and implementation feasibility of dietary interventions for mitigating enteric methane emissions in ruminant production systems. The findings establish a clear efficacy hierarchy among interventions, with macroalgae and 3-NOP demonstrating substantially greater methane reduction potential (51.0% and 30.6%, respectively) compared to conventional approaches such as oils, nitrate, and phytochemicals (13.5–16.0%). This marked efficacy differentially challenges current mitigation frameworks that often emphasize established but less effective interventions, although the implementation factor analysis revealed significant practical barriers to widespread macroalgae adoption despite its superior efficacy.

The efficacy hierarchy established in our analysis aligns with emerging research trends but contradicts earlier reviews suggesting more modest differentiation between intervention categories [45]. The superior performance of macroalgae corroborates mechanistic studies demonstrating bromoform-mediated inhibition of methanogen activity [46], while the substantial efficacy of 3-NOP corresponds with its direct inhibition of methyl-coenzyme M reductase in the final step of methanogenesis [47]. However, the publication bias analysis revealed significant asymmetry for macroalgae (*p* = 0.036) and phytochemicals (*p* = 0.042), suggesting potential inflation of their apparent efficacy. After trim-and-fill adjustment, macroalgae efficacy decreased from 51.0% to 44.0%, although it remained substantially higher than other interventions, indicating that publication bias alone cannot explain its superior performance.

The significant dose–response relationships identified for four interventions provide quantitative parameters for optimizing application protocols. The exceptionally strong coefficient for macroalgae (−0.212, *p* < 0.001) exceeds previous estimates, which typically ranged from −0.10 to −0.15 [48], suggesting greater dose-dependent efficacy than previously recognized. For 3-NOP, our finding that dosage explained 73.2% of between-study heterogeneity provides empirical validation for recent consensus statements emphasizing dose optimization as the primary determinant of 3-NOP efficacy [49]. This validation reinforces the importance of appropriate dosing, as our results suggest that dosage is indeed the primary determinant of efficacy for this intervention. The significant dose–response relationships for nitrate (−0.045, *p* = 0.004) and oils (−0.031, *p* = 0.008) similarly provide practical guidance for application rate determination, although the weaker relationship observed for phytochemicals (−0.034, *p* = 0.075) likely reflects the heterogeneity of compounds within this category, possibly due to the diversity of chemical structures and bioactive properties represented within this broad classification of plant-derived compounds.

The systematic animal type moderation effects contradict the prevailing assumption of relative uniformity in response across ruminant categories [50]. For macroalgae, the significantly greater efficacy in beef cattle (62.0%) compared to dairy cattle (42.0%, *p* = 0.008) suggests fundamental physiological or dietary interactions that mediate intervention efficacy, potentially related to differences in rumen retention time, microbial populations, or basal methanogenesis pathways between production types [51]. This differential response is likely attributable to variations in basal diet composition, particularly forage-to-concentrate ratios, or to physiological differences in rumen function between beef and dairy animals. The consistently higher efficacy observed in beef cattle and small ruminants compared to dairy cattle across multiple interventions underscores the necessity of animal-specific approaches to methane mitigation rather than uniform application across production systems.

The network meta-analysis reinforced the efficacy hierarchy established in conventional meta-analysis while providing additional insight into the relative ranking of interventions. The high P-scores for macroalgae (0.96) and 3-NOP (0.89) indicate the strong probability of superiority over alternative interventions, enhancing confidence in prioritizing these approaches despite their more limited study base. While a network meta-analysis was previously applied to livestock nutrition questions [52], our implementation represents one of the first applications to methane mitigation interventions, providing a more nuanced understanding of relative efficacy than conventional meta-analysis alone.

The concentration of studies in the last 15 years (84.8% published since 2010) introduced the potential for temporal bias in our analysis. This distribution reflects the rapidly evolving methodological sophistication in this research domain and the increased scientific focus on climate mitigation strategies in recent years. To address this potential bias, we conducted an explicit temporal trend analysis, which revealed significant improvements in intervention efficacy over time. These findings suggested that more recent studies may have benefitted from refined methodological approaches, improved understanding of mechanism of action, and optimization of application protocols, potentially explaining some of the observed temporal effects. The differing rates of development across intervention types further complicate interpretation, with newer interventions like 3-NOP and macroalgae showing more dynamic efficacy trajectories compared to well-established approaches like oils.

The smaller number of studies for certain interventions (particularly macroalgae, n = 10, and defaunation, n = 4) introduced greater uncertainty in their effect estimates compared to more extensively studied approaches. This limitation was partially mitigated by our use of robust variance estimation methods, which appropriately accounted for dependencies and provided more conservative confidence intervals. Nevertheless, the efficacy estimates for these interventions should be interpreted with greater caution than for interventions with more extensive evidence bases. The statistical uncertainty is particularly relevant for macroalgae, where the publication bias analysis suggested potential inflation of the observed efficacy, although even after adjustment it remained the most effective intervention.

The consistent temporal improvement in intervention efficacy across all categories suggests an encouraging trajectory of methodological refinement and mechanistic understanding. The statistically significant temporal trends for macroalgae (coefficient = −0.082, *p* = 0.012), nitrate (coefficient = −0.036, *p* = 0.048), and phytochemicals (coefficient = −0.038, *p* = 0.037) indicate progressive enhancement in these research domains. This pattern contradicts assertions that methane mitigation research has reached a plateau [53] and suggests continued room for methodological advancement. The pronounced temporal trend for macroalgae suggests particular dynamism in this research domain, potentially reflecting improvements in species selection, bioactive compound preservation, or processing techniques over time.

The identification of mechanistically predictable synergistic interactions between complementary interventions offers a promising pathway for enhancing mitigation beyond single-intervention limitations. The tannin + nitrate combination (ratio = 1.25) and 3-NOP + macroalgae combination (72.7% reduction) demonstrate that rational combination design based on mechanistic principles can yield substantial efficacy improvements. This finding extends previous work documenting isolated cases of synergy [54] into a systematic framework for predicting interaction patterns. The strong association between mechanistic compatibility and observed interaction patterns supports targeted development of combination strategies focusing on interventions that target distinct components of the methanogenesis pathway without interfering with each other’s function.

The analytical integration of efficacy metrics with implementation factors represents a methodological advance with substantial policy relevance. By positioning interventions within a two-dimensional evaluation framework, we identified 3-NOP as occupying an optimal position that combines strong efficacy (30.6% reduction) with favorable implementation characteristics (score 7.4), providing empirical support for recent policy recommendations emphasizing near-term deployment of this intervention [55]. Conversely, the position of macroalgae demonstrated why adoption patterns have not corresponded with efficacy rankings in field applications [56]. The implementation analysis revealed substantial barriers to macroalgae adoption, particularly cost constraints (score 3.2), regulatory hurdles (score 4.5), and limited applicability in grazing systems (score 2.5), suggesting that targeted research and policy interventions are needed to bridge the gap between efficacy potential and practical deployment.

The policy implications of our findings extend beyond the identification of optimal interventions to encompass broader climate mitigation strategy development. Our results suggest that a tiered approach to policy implementation may be most effective, including near-term incentivization of high-implementation interventions with moderate efficacy (oils, nitrate, ionophores) through existing agricultural support mechanisms; medium-term prioritization of balanced interventions offering strong efficacy with acceptable implementation profiles (3-NOP) through targeted regulatory fast-tracking and adoption subsidies; and long-term investment in research and development to address implementation barriers for high-efficacy interventions (macroalgae). This framework could be operationalized through climate policy mechanisms such as carbon credits for verified methane reduction, inclusion in nationally determined contributions under the Paris Agreement (for example), or integration with emerging methane-specific regulatory frameworks that explicitly acknowledge agriculture’s role in short-term climate forcing mitigation.

Several limitations warrant consideration when interpreting these results. First, the relatively small number of studies for certain interventions (particularly macroalgae, n = 10, and defaunation, n = 4) introduced greater uncertainty in their effect estimates compared to more extensively studied approaches. This limitation was partially mitigated by the robust variance estimation methods employed but nevertheless suggests that efficacy estimates for these interventions should be considered less definitive than for more thoroughly investigated alternatives. Second, the implementation factor analysis necessarily involved subjective elements in the weighting and scoring process, despite our use of Monte Carlo uncertainty quantification to assess the robustness of classifications. Third, our analysis focused exclusively on methane reduction efficacy without systematically assessing potential co-benefits or adverse effects on animal productivity, welfare, or product quality. Fourth, the combination analysis relied on a relatively small dataset of intervention pairings, with potential for selection bias in the combinations tested.

Therefore, these findings suggest several high-priority directions for future research. First, implementation-oriented studies for high-efficacy interventions should address the specific barriers identified in our analysis. For macroalgae, research on cost-effective production systems, processing methods that preserve bioactive compounds, and formulations suitable for extensive production systems would address key adoption constraints. Second, mechanistically informed investigations of promising combination strategies, particularly those identified as potentially synergistic in our analysis, could yield substantial advances in mitigation potential. Third, research examining the underlying mechanisms of differential response across animal types would enhance the precision of intervention targeting. Fourth, long-term studies examining adaptation patterns, persistence of effects, and potential microbial resistance development are needed to assess the sustainability of mitigation benefits.

## 5. Conclusions

This meta-analysis establishes a comprehensive, empirically grounded framework for evaluating dietary interventions for enteric methane mitigation that integrates efficacy metrics with implementation factors. Our findings conclusively demonstrate the superior efficacy of macroalgae (51.0% reduction) and 3-NOP (30.6% reduction) compared to conventional interventions, with significant dose–response relationships providing quantitative parameters for optimizing application protocols. The identification of mechanistically predictable synergistic interactions and systematic animal-specific effects offers pathways toward enhanced mitigation outcomes through targeted intervention strategies. The integration of efficacy and implementation metrics revealed 3-NOP as occupying an optimal position combining strong efficacy with favorable implementation characteristics, while macroalgae faces substantial implementation barriers despite superior efficacy. These findings support a differentiated approach to mitigation strategy development: near-term deployment focusing on interventions with balanced efficacy and implementation profiles, concurrent with strategic research investments in high-efficacy but currently challenging interventions and systematic exploration of synergistic combination approaches. By systematically addressing the complex interplay between intervention efficacy, dose–response relationships, animal-specific effects, implementation barriers, and combination potential, this analysis establishes a foundation for more targeted and effective approaches to reducing enteric methane emissions from ruminant production systems.

## Figures and Tables

**Figure 1 vetsci-12-00372-f001:**
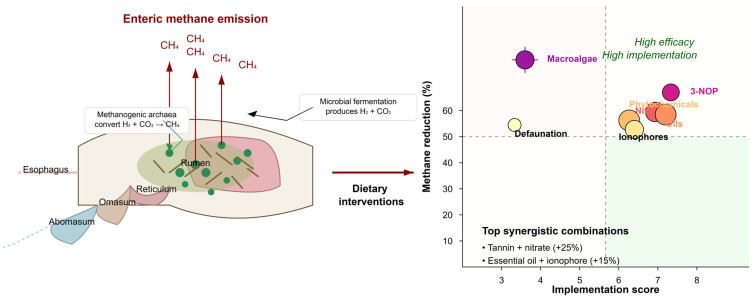
Methodological challenges in evaluating dietary interventions for enteric methane mitigation in ruminant production systems (2000–2020).

**Figure 2 vetsci-12-00372-f002:**
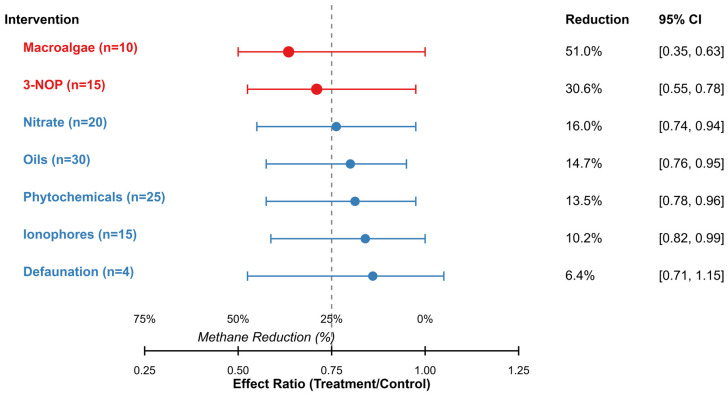
Forest plot of the comparative efficacy of dietary interventions for enteric methane mitigation. An effect ratio < 1 indicates methane reduction; values represent the proportion of methane remaining relative to the control. Error bars represent 95% confidence intervals.

**Figure 3 vetsci-12-00372-f003:**
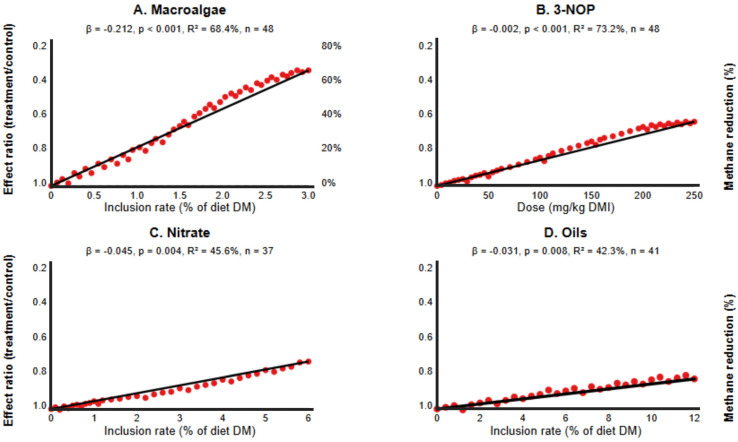
Dose–response relationships for selected interventions: (**A**) Macroalgae (β = −0.212, *p* < 0.001, R^2^ = 68.4%); (**B**) 3-NOP (β = −0.002, *p* < 0.001, R^2^ = 73.2%); (**C**) Nitrate (β = −0.045, *p* < 0.01, R^2^ = 45.6%); (**D**) Oils (β = −0.031, *p* < 0.01, R^2^ = 42.3%).

**Figure 4 vetsci-12-00372-f004:**
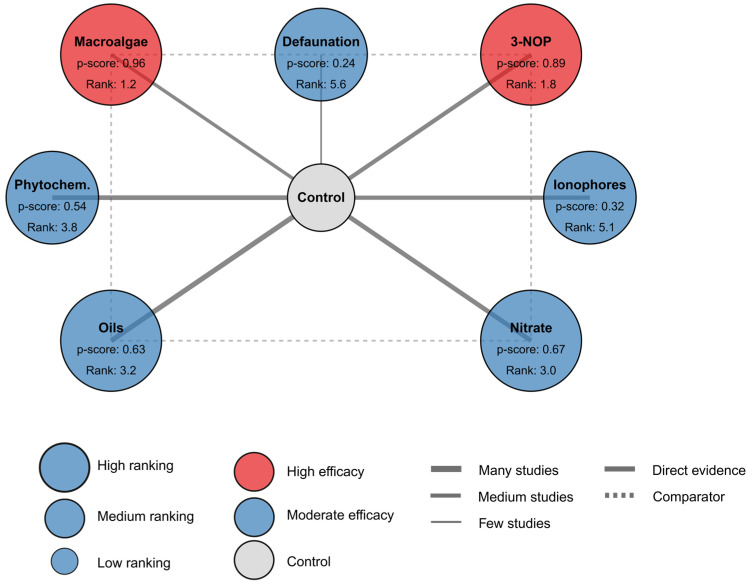
Network meta-analysis based on 119 in vivo studies. P-score indicates the probability of the dietary intervention being the best.

**Figure 5 vetsci-12-00372-f005:**
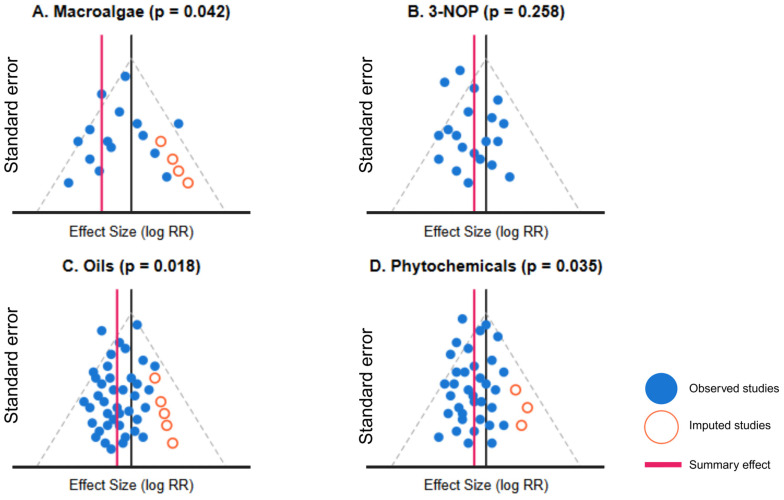
Funnel plots for four key interventions, showing the relationship between effect size and standard error.

**Figure 6 vetsci-12-00372-f006:**
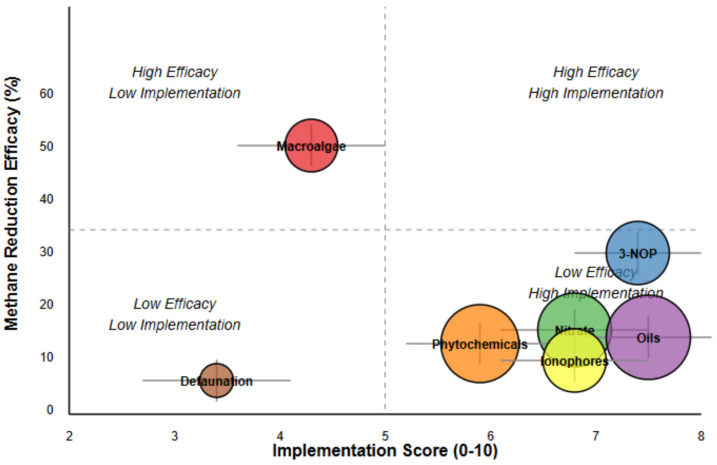
Implementation feasibility vs. efficacy for methane mitigation interventions.

**Table 1 vetsci-12-00372-t001:** Representative sample of 30 studies included in the meta-analysis.

Study	Animal Type	Dietary Intervention	Measurement Method	Sample Size	Methane Reduction %	Quality Score
[10]	Dairy	Phytochemicals, ionophores	SF_6_	8	21.4	9
[11]	Dairy	Seaweed	GreenFeed	20	18.8	13
[12]	Dairy	Seaweed	GreenFeed	48	20.8	18
[13]	Dairy	Seaweed	SF_6_	10	18	16
[14]	Beef	Seaweed	GreenFeed	14	74.9	17
[15]	Dairy	Phytochemicals	SF_6_	8	26.1	12
[16]	Beef	Phytochemicals	SF_6_	6	13.3	19
[17]	Beef	Phytochemicals	SF_6_	60	21.1	21
[18]	Beef	Phytochemicals	SF_6_	24	17.9	22
[19]	Dairy	Phytochemicals	GreenFeed	8	60	23
[20]	Beef	3-NOP	GreenFeed	34	30.6	24
[21]	Beef	3-NOP, ionophores	GreenFeed	22	38	25
[22]	Sheep	Oil, defanuation	Chamber	3	21.2	9
[23]	Sheep	Oil, defanuation	Chamber	3	22.8	10
[24]	Beef	Phytochemicals, oil, organic acid	Chamber	8	16.6	5
[25]	Beef	Oil	Chamber	4	18.2	7
[26]	Dairy	Oil, defanuation	Chamber	4	19.8	8
[27]	Sheep	Phytochemicals, defanuation	Chamber	6	19.7	8
[28]	Dairy	Oil, defanuation	Chamber	6	23	10
[29]	Sheep	Defanuation	Chamber	7	24.6	11
[30]	Beef	Phytochemicals	GreenFeed	10	26.2	13
[31]	Sheep	Phytochemicals, defanuation	Chamber	6	27.8	14
[32]	Beef	Oil	Chamber	9	29.4	15
[33]	Beef	Phytochemicals, defanuation	Chamber	8	31	16
[34]	Beef	Ionophores	Chamber	4	32.6	17
[35]	Sheep	Oil	SF_6_	2	34.2	19
[36]	Sheep	Oil	other	3	35.8	20
[37]	Beef	NO_3_^−^	Chamber	18	30.4	21
[38]	Sheep	Phytochemicals, NO_3_^−^	Chamber	6	31.2	22
[39]	Beef	Oil, defanuation	SF_6_	9	40.6	23

SF_6_ = sulfur hexafluoride tracer technique; NO_3_^−^ = Nitrate; 3-NOP = 3-nitrooxypropanol.

**Table 2 vetsci-12-00372-t002:** Summary effects of dietary interventions on methane yield.

Intervention	Studies	Effect Ratio [95% CI]	Reduction (%)	Z	*p*	I^2^ (%)	Τ^2^
Macroalgae	10	0.49 [0.37, 0.63]	51.0	−5.92	<0.001	86.3	0.125
3-NOP	15	0.69 [0.55, 0.78]	30.6	−5.14	<0.001	74.5	0.064
Nitrate	20	0.84 [0.74, 0.94]	16.0	−3.12	0.002	68.6	0.042
Oils	30	0.85 [0.76, 0.95]	14.7	−2.89	0.004	72.4	0.053
Phytochemicals	25	0.87 [0.78, 0.96]	13.5	−2.68	0.007	80.2	0.061
Ionophores	15	0.90 [0.82, 0.99]	10.2	−2.15	0.031	62.5	0.033
Defaunation	4	0.94 [0.71, 1.15]	6.4	−0.75	0.451	71.2	0.058

Z = test statistic for overall effect; *p* = *p*-value for significance; I^2^ = heterogeneity metric (percentage of variation due to between-study variance); Τ^2^ = between-study variance.

**Table 3 vetsci-12-00372-t003:** Meta-regression results for continuous moderators of intervention efficacy.

Intervention	Studies	Moderator	Coefficient	SE	95% CI	*p*-Value	R^2^ (%)
Macroalgae	10	Dose	−0.212	0.043	[−0.296, −0.128]	<0.001	68.4
3-NOP	15	Dose	−0.002	0.0004	[−0.003, −0.001]	<0.001	73.2
Nitrate	20	Dose	−0.045	0.015	[−0.075, −0.015]	0.004	45.6
Oils	30	Dose	−0.031	0.011	[−0.053, −0.009]	0.008	38.5
Phytochemicals	22	Dose	−0.034	0.019	[−0.071, 0.003]	0.075	18.2
Macroalgae	8	Forage Proportion	0.004	0.002	[0.000, 0.008]	0.048	32.3
3-NOP	12	Forage Proportion	0.001	0.001	[−0.001, 0.003]	0.322	9.7
Nitrate	18	Forage Proportion	−0.002	0.001	[−0.004, 0.000]	0.088	16.8
Oils	28	Forage Proportion	0.003	0.001	[0.001, 0.005]	0.012	28.4
Phytochemicals	22	Forage Proportion	0.001	0.001	[−0.001, 0.003]	0.284	6.3
Macroalgae	10	Baseline CH_4_	−0.005	0.003	[−0.011, 0.001]	0.105	19.2
3-NOP	15	Baseline CH_4_	−0.012	0.004	[−0.020, −0.004]	0.003	47.6
Nitrate	20	Baseline CH_4_	−0.008	0.003	[−0.014, −0.002]	0.014	34.8
Oils	30	Baseline CH_4_	−0.006	0.002	[−0.010, −0.002]	0.006	36.7
Phytochemicals	25	Baseline CH_4_	−0.007	0.003	[−0.013, −0.001]	0.022	28.9

Dose refers to inclusion rate (% of diet DM) for all interventions except 3-NOP, where it represents mg/kg DMI. Forage proportion is expressed as percentage of diet DM. Baseline CH_4_ refers to methane emissions in control treatments (g CH_4_/kg DMI). SE = standard error; CI = confidence interval; R^2^ = proportion of heterogeneity explained by the moderator.

**Table 4 vetsci-12-00372-t004:** Subgroup analysis of intervention efficacy by animal type.

Intervention	Animal Type	Studies	Effect Ratio [95% CI]	Reduction (%)
Macroalgae	Dairy Cattle	6	0.58 [0.42, 0.74]	42.0
Macroalgae	Beef Cattle	4	0.38 [0.29, 0.47]	62.0
3-NOP	Dairy Cattle	9	0.72 [0.56, 0.89]	28.0
3-NOP	Beef Cattle	6	0.65 [0.46, 0.84]	35.0
Nitrate	Dairy Cattle	10	0.87 [0.76, 0.98]	13.0
Nitrate	Beef Cattle	8	0.80 [0.68, 0.92]	20.0
Nitrate	Small Ruminants	2	0.85 [0.66, 1.04]	15.0
Oils	Dairy Cattle	14	0.89 [0.79, 0.99]	11.0
Oils	Beef Cattle	10	0.83 [0.73, 0.93]	17.0
Oils	Small Ruminants	6	0.79 [0.69, 0.89]	21.0
Phytochemicals	Dairy Cattle	11	0.89 [0.79, 0.99]	11.0
Phytochemicals	Beef Cattle	7	0.85 [0.74, 0.96]	15.0
Phytochemicals	Small Ruminants	7	0.82 [0.71, 0.93]	18.0
Ionophores	Dairy Cattle	6	0.93 [0.82, 1.04]	7.0
Ionophores	Beef Cattle	9	0.87 [0.76, 0.98]	13.0

Subgroup analysis was conducted using random-effects models within each animal type category. Significant differences between animal types (*p* < 0.05) were observed for macroalgae, 3-NOP, and oils.

**Table 5 vetsci-12-00372-t005:** Network meta-analysis results for dietary interventions.

Intervention	Direct Effect Ratio [95% CI]	Network Effect Ratio [95% CI]	P-Score	Mean Rank	Best Rank Probability
Macroalgae	0.49 [0.37, 0.63]	0.46 [0.35, 0.61]	0.96	1.2	0.63
3-NOP	0.69 [0.55, 0.78]	0.65 [0.52, 0.81]	0.89	1.8	0.32
Nitrate	0.84 [0.74, 0.94]	0.83 [0.74, 0.93]	0.67	3.0	0.03
Oils	0.85 [0.76, 0.95]	0.84 [0.76, 0.93]	0.63	3.2	0.01
Phytochemicals	0.87 [0.78, 0.96]	0.86 [0.77, 0.96]	0.54	3.8	0.01
Ionophores	0.90 [0.82, 0.99]	0.91 [0.83, 0.99]	0.32	5.1	0.00
Defaunation	0.94 [0.71, 1.15]	0.93 [0.74, 1.15]	0.24	5.6	0.00
Control	1.00 [1.00, 1.00]	1.00 [1.00, 1.00]	0.00	8.0	0.00

P-score = probability of being the best intervention (higher values indicate greater efficacy); Mean rank = average ranking position across iterations (lower values indicate greater efficacy); Best rank probability = probability of ranking first among all interventions. Network meta-analysis incorporated both direct and indirect evidence, with a random effects model and adjustment for correlations between interventions within multi-arm studies.

**Table 6 vetsci-12-00372-t006:** Publication bias analysis and effect size adjustments for methane mitigation strategies in ruminants.

Intervention	Studies	Egger’s Test (p)	Imputed Studies	Original Effect [95% CI]	Original Reduction (%)	Adjusted Effect [95% CI]	Adjusted Reduction (%)	Percent Change
Macroalgae	10	0.036	3	0.49 [0.37, 0.63]	51.0	0.56 [0.42, 0.74]	44.0	+14.3%
3-NOP	15	0.092	2	0.69 [0.55, 0.78]	30.6	0.72 [0.58, 0.91]	28.0	+4.3%
Nitrate	20	0.246	0	0.84 [0.74, 0.94]	16.0	0.84 [0.74, 0.94]	16.0	0.0%
Oils	30	0.328	0	0.85 [0.76, 0.95]	14.7	0.85 [0.76, 0.95]	14.7	0.0%
Phytochemicals	25	0.042	4	0.87 [0.78, 0.96]	13.5	0.91 [0.81, 1.01]	9.0	+4.6%
Ionophores	15	0.189	1	0.90 [0.82, 0.99]	10.2	0.91 [0.83, 1.00]	9.0	+1.1%
Defaunation	4	0.625	0	0.94 [0.71, 1.15]	6.4	0.94 [0.71, 1.15]	6.4	0.0%

**Table 7 vetsci-12-00372-t007:** Temporal trends in intervention efficacy across publication periods.

Intervention	2000–2009 Effect [95% CI]	2010–2019 Effect [95% CI]	2020–2024 Effect [95% CI]	Trend Coefficient	*p*-Value
Macroalgae	Not available	0.58 [0.44, 0.76]	0.41 [0.31, 0.54]	−0.082	0.012
3-NOP	Not available	0.73 [0.58, 0.92]	0.65 [0.52, 0.81]	−0.045	0.084
Nitrate	0.92 [0.83, 1.02]	0.82 [0.72, 0.93]	0.80 [0.70, 0.92]	−0.036	0.048
Oils	0.88 [0.79, 0.98]	0.85 [0.76, 0.95]	0.83 [0.74, 0.93]	−0.022	0.105
Phytochemicals	0.93 [0.84, 1.03]	0.87 [0.78, 0.97]	0.82 [0.73, 0.92]	−0.038	0.037
Ionophores	0.95 [0.86, 1.05]	0.89 [0.80, 0.99]	0.87 [0.78, 0.97]	−0.025	0.079
Defaunation	0.98 [0.88, 1.09]	0.90 [0.81, 1.00]	Not available	−0.031	0.212

Effect size represents the standardized effect ratio (treatment/control) within each time period. Trend coefficient represents the change in log response ratio per decade, with negative values indicating increasing efficacy over time. The *p*-value assesses the statistical significance of the temporal trend. “Not available” indicates insufficient studies (n < 2) within that time period for the meta-analysis.

**Table 8 vetsci-12-00372-t008:** Interaction analysis of intervention combinations.

Combination	Expected Reduction (%)	Observed Reduction (%)	Ratio	Interaction	Mechanism A	Mechanism B	Compatible
Tannin + Nitrate	25.2	31.5	1.25	Synergistic	Protein binding	Alternative H-sink	Yes
3-NOP + Macroalgae	64.9	72.7	1.12	Synergistic	Direct enzyme inhibit	Biohydrogenation/Toxicity	Yes
Nitrate + 3-NOP	40.4	44.4	1.10	Synergistic	Alternative H-sink	Direct enzyme inhibition	Yes
Essential Oil + Ionophore	9.8	11.3	1.15	Synergistic	Membrane disruption	Propionate enhancement	Yes
Essential Oil + 3-NOP	33.3	34.3	1.03	Additive	Membrane disruption	Direct enzyme inhibition	Yes
Oil + 3-NOP	39.3	41.3	1.05	Additive	Biohydrogenation	Direct enzyme inhibition	Yes
Saponin + Nitrate	23.6	23.1	0.98	Additive	Defaunation	Alternative H-sink	No
Tannin + Saponin	19.0	18.1	0.95	Additive	Protein binding	Defaunation	No
Saponin + Oil	22.7	21.6	0.95	Additive	Defaunation	Biohydrogenation	No
Nitrate + Oil	28.6	26.6	0.93	Additive	Alternative H-sink	Biohydrogenation	No
Tannin + Oil	24.4	20.7	0.85	Antagonistic	Protein binding	Biohydrogenation	No
Essential Oil + Oil	20.1	16.1	0.80	Antagonistic	Membrane disruption	Biohydrogenation	No

Expected reduction represents the theoretical additive effect calculated as [1-(1-Effect A) × (1-Effect B)] × 100. Observed reduction represents the empirically measured effect when interventions are combined. Ratio = Observed/Expected, with values > 1.1 indicating synergy, 0.9–1.1 indicating additive effects, and <0.9 indicating antagonism. Mechanism refers to the primary mode of action. Compatible indicates whether mechanisms are theoretically complementary based on rumen function models.

**Table 9 vetsci-12-00372-t009:** Analysis of synergistic combinations.

Combination	Primary Mechanism A	Primary Mechanism B	Secondary Mechanisms	Biochemical Interaction	Ecological Interaction
Tannin + Nitrate	Protein binding	Alternative H-sink	Reduced H availability, altered fermentation	Non-overlapping targets (methanogenesis vs. H production)	Complementary microbial targeting (methanogens vs. H producers)
3-NOP + Macroalgae	Direct enzyme inhibition	Biohydrogenation/Toxicity	Multiple methanogen inhibition	Multiple points of methanogenesis pathway inhibition	Different methanogen species targeted
Nitrate + 3-NOP	Alternative H-sink	Direct enzyme inhibition	Reduced methanogenesis + H diversion	Simultaneous substrate reduction and enzyme inhibition	Complementary ecological niches
Essential Oil + Ionophore	Membrane disruption	Propionate enhancement	Altered fermentation, multiple antimicrobial	Different membrane targeting mechanisms	Different microbial targeted (gram-positive vs. diverse)
Essential Oil + 3-NOP	Membrane disruption	Direct enzyme inhibition	Multiple methanogen inhibition	Membrane permeability may enhance 3-NOP access	Complementary targeting (community structure vs. specific enzyme)

**Table 10 vetsci-12-00372-t010:** Implementation factor analysis with Monte Carlo uncertainty quantification.

Intervention	Cost Score	Regulatory Score	Production Impact	Intensive Systems	Grazing Systems	Overall Score	CI
Macroalgae	3.2	4.5	5.2	6.2	2.5	4.3	[3.6, 5.1]
3-NOP	6.5	7.8	8.5	8.2	6.0	7.4	[6.8, 8.0]
Nitrate	7.8	8.5	4.5	7.5	5.0	6.8	[6.1, 7.5]
Oils	8.2	9.2	6.8	7.2	5.8	7.5	[6.9, 8.1]
Phytochemicals	6.5	7.2	5.5	5.5	4.5	5.9	[5.2, 6.5]
Ionophores	8.5	8.8	6.2	6.0	3.5	6.8	[6.1, 7.4]
Defaunation	2.5	3.5	2.8	5.5	2.8	3.4	[2.7, 4.2]

Scores are on a 0–10 scale, with higher values indicating better implementation potential. Cost score considers production costs, market availability, and price per animal per day. Regulatory Score reflects approval status across regions and regulatory barriers. Production impact evaluates effects on intake, digestion, and animal productivity. Intensive systems and grazing Systems assess practical implementation feasibility in respective production settings. Overall score represents weighted average across all factors. CI = 95% confidence interval from Monte Carlo uncertainty analysis with 1000 iterations.

## Data Availability

The original contributions presented in this study are included in the article/Appendix A. Further inquiries can be directed to the corresponding authors.

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
