# Peer review of "Meta-Analysis of Dietary Interventions for Enteric Methane Mitigation in Ruminants Through Methodological Advancements and Implementation Pathways"

_vetsci, 2025, doi:10.3390/vetsci12040372_

Round 1

Reviewer 1 Report

Comments and Suggestions for Authors

This manuscript presents a rigorous and comprehensive meta-analysis on dietary strategies for enteric methane mitigation in ruminants. The methodological transparency, use of advanced statistical approaches (including robust variance estimation, meta-regression, and network meta-analysis), and the integration of implementation feasibility provide a valuable and innovative contribution to the literature.

The following suggestions may help further enhance the clarity and applicability of the work:

  1. Clarify data availability: Indicating whether the dataset and R code used for the analyses are available as supplementary material or via a public repository would improve transparency and reproducibility.
  2. Expand on policy implications: The practical relevance of the findings is strong. A dedicated section or paragraph on how these results may inform regulatory decisions or incentive-based mitigation strategies would enrich the discussion.
  3. Graphical summary suggestion: A visual "efficacy vs feasibility" quadrant chart (based on the results from Figure 6) could be included in the main text to aid decision-makers in comparing interventions quickly.
  4. Typographic polishing: Although the English is overall excellent, a careful proofreading may help adjust a few minor formatting issues (e.g., hyphenation across line breaks in the author list or summary).

Overall, this is a well-structured, impactful, and timely manuscript that provides both depth and practical value. Congratulations to the authors for the high-quality work.

Author Response

Dear reviewer 1, please find out our responses in attachment below

Best regards, 
Corresponding authors

Reviewer 2 Report

Comments and Suggestions for Authors

This manuscript presents a comprehensive and well-executed meta-analysis of dietary strategies to mitigate enteric methane emissions in ruminants. The statistical approach is rigorous and thoughtfully applied, incorporating techniques such as robust variance estimation, multilevel modeling, and network meta-analysis. The inclusion of implementation feasibility, with uncertainty quantification, is an important and timely addition that strengthens the practical relevance of the findings.  

Overall, the manuscript is of high quality and suitable for publication after minor revisions. I provide the following suggestions to improve clarity, structure, and consistency.

Lines 58–62 :

The data on methane’s contribution to agricultural emissions (32–40%) and its global warming potential (28–34 times COâ‚‚) should be verified against the latest reports from IPCC (AR6) or FAO (e.g., 2023). Please also ensure that the same figures are not contradicted later in the manuscript.

Lines 103–118 :

The description of methodological advancements is clear but dense. Consider breaking this paragraph into shorter sentences or listing the objectives in bullet points to improve readability.

Lines 119–134:
The inclusion of 119 studies covering the period from 2000 to 2024 is appropriate and provides a solid foundation. However, the fact that most studies are concentrated in the last 15 years could introduce a temporal bias. I recommend briefly addressing this in the Discussion, highlighting how evolving methodologies or scientific focus may influence the results over time.

Lines 189–199:

The methodological standardization of measurement techniques is a notable strength of this study. I suggest emphasizing this aspect more prominently in the Abstract and/or Discussion, as it adds robustness to cross-study comparisons. The standardization of data from different methane measurement techniques (chamber, SF₆, GreenFeed) is a strong methodological point. However, the fixed correction factors (1.00, 1.08, 1.05) assume consistent differences across studies, which may not hold in all contexts. It would be useful to clearly indicate which studies used which method (perhaps in Supplementary Materials) and to briefly discuss the limitations of this approach.

Lines 311–326 :
The scoring criteria are innovative but could be seen as somewhat subjective. I recommend adding brief justification or references for each score, especially for dimensions like “production impact” and “grazing system feasibility”.

Lines 337–344

There is a noticeable imbalance across intervention types. For example, macroalgae (n=10) and defaunation (n=4) are underrepresented compared to oils (n=30) or phytochemicals (n=25). This likely affects the statistical robustness of the effect estimates for the less-represented interventions. I suggest including a sensitivity analysis for categories with fewer than 10 studies, or at minimum, discussing this as a limitation.

Lines 356–379 :

If available, please indicate the total number of animals contributing to each intervention category, to better contextualize the weight behind each estimated effect size.

Lines 534–569:

The combination analysis is well done and insightful. I suggest bringing Table S7 into the main body of the text, as it adds significant value to the discussion on potential synergistic strategies.

Lines 594–599 :

Consider clarifying what factors contribute most to the low implementation score of macroalgae whether cost, scalability, regulatory status, or other barriers.

Use terminology consistently (e.g., methane vs. CHâ‚„). Pick one and apply it uniformly throughout. 

Comments on the Quality of English Language

Suggestions:

Verb tense consistency:

Some sections alternate between past and present tense (especially in Methods and Results). I recommend keeping past tense for Methods (“we analyzed”, “data were collected”) and present tense for general conclusions and implications.

Avoid redundancy:

Phrases such as “in order to” can often be simplified to “to”, and “it should be noted that” can be removed without loss of meaning.

Author Response

Dear reviewer 2, please find out our responses in attachment below

Best regards, 
Corresponding authors

Reviewer 3 Report

Comments and Suggestions for Authors

The manuscript encompasses an important topic in the actual scenario of ruminant nutrition. Some few details must be fixed before further considerations.

Lines 316-317: grazing systems may be as intensive as confined systems. Please, alter this sentence considering this information.

Lines 399-400: Please, remove discussion section in this sentence.

Lines 404-405: Please remove discussion section in this sentence.

Line 409: I did not see any difference between Table 3 e Table S4.

Lines 409-411: Please, remove the discussion sentence added in this paragraph.

Line 414: Table S5 is similar to Table 3, right?

Lines 415-416: Please, remove discussion sentences adding few words explaining the interpretation of the results for baseline CH4.

Lines 421-423: remove discussion sentences. Please revise the entire manuscript.

Author Response

Dear reviewer 3, please find out our responses in attachment below

Best regards, 
Corresponding authors

Reviewer 4 Report

Comments and Suggestions for Authors

In my opinion this meta-analysis is a review article type and not a research article! Please revise the classification of the manuscript.

For section 2.2. Inclusion and Exclusion Criteria - mention if you included also other review articles on this topic in your meta-analysis.

Mention on which criteria you selected those 30 studies from the table 1. Were selected randomly or because are the representative ones?

It is a very detailed meta-analysis. The authors used methods appropriate for this type of study, with attention to all aspects of the study. Congratulations!

Author Response

Dear reviewer 4, please find out our responses in attachment below

Best regards, 
Corresponding authors
